# On-Resin Selenopeptide Catalysts: Synthesis and Applications of Enzyme-Mimetic Reactions and Cyclization of Unsaturated Carboxylic Acids [note 1]

**DOI:** 10.3390/molecules30030480

**Published:** 2025-01-22

**Authors:** Michio Iwaoka, Yua Maese, Kasumi Abe

**Affiliations:** 1Department of Chemistry, School of Science, Tokai University, Kitakaname, Hiratsuka-shi 259-1292, Kanagawa, Japan; 2Institute of Advanced Biosciences, Tokai University, Kitakaname, Hiratsuka-shi 259-1292, Kanagawa, Japan

**Keywords:** selenocysteine, solid-phase peptide synthesis, green chemistry, redox reaction, glutathione peroxidase, NH···Se hydrogen bond, SAAP simulation, electron probe micro analyzer (EPMA)

## Abstract

Selenium reagents are useful for selenoenzyme-mimicking reactions, as well as for organic synthesis. However, the reaction waste containing selenium frequently smells unpleasant and exhibits serious toxicity. Herein, we have developed new-type on-resin selenium reagents, H-UXX···-PAM (**5**) and Ac-(X)U*XX···-PAM (**6**), where U and U* represent selenocysteine (U) and *p*-methoxybenzyl (PMB)-protected U, respectively, as recyclable catalysts, in which U-containing peptide chains are linked to the polystyrene resin PAM. Synthesized on-resin selenopeptides **5a**–**g** with a variable amino acid sequence were evaluated for their glutathione peroxidase (GPx)-like activity using the UV and ^1^H NMR methods, using the reaction between dithiothreitol (DTT^red^) and H_2_O_2_ in methanol. It was found that the intramolecular interaction between U and a basic amino acid residue, such as histidine (H) and lysine (K), enhances peroxidase activity through the formation of an NH···Se hydrogen bond. On the other hand, the catalytic activity of **6a**–**d** was evaluated in the oxidative cyclization of β,γ-unsaturated acids (**7**) into α,β-unsaturated lactones (**8**). Although the yield of **8** was significantly decreased after second- or third-round reaction, due to detachment of the selenium moiety from the resin, the results demonstrated reusability, as well as a substrate scope of **6** as a catalyst. Since U is a natural amino acid, on-resin selenopeptides are potential targets as novel-type green redox catalysts.

## 1. Introduction

Selenium has characteristic chemical and biological properties, as it is easily reducible and oxidizable to multiple oxidation states [1]. For example, the reduced selenol form, i.e., RSeH, is not only a good nucleophile to various organic substrates, but also an efficient redox catalyst against reactive oxygen species, such as hydrogen peroxide (H_2_O_2_) [2,3,4]. Taking advantage of these unique features, a number of synthetic reactions using organoselenium compounds have been developed [5,6,7,8,9]. However, those utilizing L-selenocysteine (Sec, U), which is a unique natural amino acid existing at the active sites of various selenoenzymes, such as glutathione peroxidase (GPx) and thioredoxin reductase (TrxR), are poorly known [10,11,12]. In our laboratory, various Sec derivatives have been synthesized for the use of chemical biology studies as well as organic synthesis [13,14]. For example, Sec-containing peptides (i.e., selenopeptides) were designed to mimic the local active site structures of GPx and TrxR [15,16,17,18]. α-Methyl-L-selenocysteine derivatives with adequate protecting groups were also synthesized as asymmetric catalysts for the cyclization reaction of unsaturated carboxylic acids, though only low asymmetric induction was achieved [19].

In the meantime, the reaction waste containing selenium frequently smells unpleasant and exhibits serious toxicity [20,21,22]. To reduce these environmental risks, heterogeneous selenium reagents, in which the reaction center Se atom is bound to a polymer support, have been developed elaborately since the late 1990s (Figure 1a) [23,24,25]. Ruhland [26] devised a polystyrene-bound nucleophilic selenium reagent (**1**), while Nicolaou [27,28,29] and Fujita [30,31,32] synthesized electrophilic alternatives (**2–4**). Wirth also developed chiral electrophilic reagents [33]. These pioneering works have stimulated many researchers to develop various on-resin selenium reagents for the synthesis of a wide range of organic compounds, such as functionalized alkenes [34,35,36], dihydrofurans [37,38], triazoles [39,40,41,42], flavonoids [43], and so forth [44,45,46]. However, most reagents were for stoichiometric use only, except for particular cases. Barrero et al. [47] previously applied reagent **3** to allylic chlorination as a catalyst (Figure 1b). Recently, Tan et al. [48] reported interesting microgels, which catalyzed the conversion of acrolein into acrylic acid, with H_2_O_2_ mediated by the polymer-bound seleninic acid moiety (RSeO_2_H) (Figure 1c). In this context, we have developed new-type on-resin selenium reagents (**5** and **6**) (Table 1) as recyclable catalysts, in which selenopeptide chains are linked to the polystyrene resin surface. In this study, these catalysts, with variable amino acid sequences and protecting groups, were evaluated for their GPx-like peroxidase activities, as well as their reusability in the catalytic conversion of β,γ-unsaturated acids (**7**) into α,β-unsaturated lactones (**8**).

## 2. Results and Discussion

### 2.1. Synthesis of On-Resin Selenopeptide Catalysts

Catalysts **5** and **6** were synthesized according to the scheme shown in Figure 2. As the resin base, (4-hydroxymethylphenyl)acetamidomethyl polystyrene (PAM), which is usually used for the Boc method peptide synthesis [49,50], was employed. After deprotection of the Boc group of Boc-Leu-PAM or Boc-Gly-PAM with trifluoroacetic acid (TFA), amino acids were sequentially connected to the C-terminal amino acid, i.e., Leu (L) or Gly (G), on the resin, following common Fmoc method protocols. The Sec (U) residue was attached to the peptide using Fmoc-Sec(PMB)-OH (PMB: *p*-methoxybenzyl). After elongation of the peptide chain, the Fmoc group on the N-terminal residue was deprotected with piperidine. To obtain **5a**–**g**, the resulting resin was treated with a TFA cocktail containing 2,2′-dipyridyl disulfide (DPDS), triisopropylsilane (TIS), and H_2_O, and then with dithiothreitol (DTT^red^) to deprotect all protecting groups on the sidechains. On the other hand, **6a**–**d** were obtained by capping the N-terminus with an acetyl group, keeping the PMB protection on the Sec residue. The purity and identity of the selenopeptides grown on the resin were confirmed by cleaving the peptides from the resin with trifluoromethanesulfonic acid (TfOH), followed by HPLC and MALDI-TOF-MS analyses (see Experimental section and Appendix A). The yields of the peptides isolated as a diselenide form were 11–57% based on the amount of the first amino acid preloaded on the PAM resin. The low yields were due to several factors, such as a loss during the isolation process by HPLC or the formation of other selenium species, such as a monomeric selenol or an ethanedithiol (EDT) adduct, during the peptide cleavage process. EDT was added to the solution to promote the cleavage reaction. It is of note that other peptide species were not detected as major peaks on the HPLC charts of the cleaved peptide mixtures, indicating that the selenopeptides were successfully synthesized on the resin.

### 2.2. GPx-like Peroxidase Activity of ***5***

The GPx-like peroxidase activity of catalyst **5** was evaluated in the reaction between H_2_O_2_ and DTT^red^ in methanol (Figure 3a and Appendix A). This reaction is frequently used as a model reaction for a GPx-like or TrxR-like activity assay [51,52]. We recently reported that the TrxR-like antioxidant activity of selenopeptide H-CUGHGE-OH is enhanced by the intramolecular NH···Se hydrogen bond formed between the basic His (H) sidechain and the Se atom of the Sec (U) residue [14]. According to this observation, we designed a series of on-resin catalysts with the amino acid sequences of H-UA*_n_*HGEL-PAM (*n* = 0, 1, 3, and 4) (**5a**–**d**) using Boc-Leu-PAM as a resin base. As the reaction proceeded, an increase in the UV absorption at 310 nm was observed, due to the formation of oxidized dithiothreitol (DTT^ox^) (Figure 3b). It is notable that the catalytic activity decreased with the number of Ala (A) spacers, suggesting that the GPx-like peroxidase activity was enhanced by the interaction between U and H. Subsequently, the His (H) residue was replaced by Lys (K) or Pro (P) fixing *n* = 1. The obtained H-UAKGEL-PAM (**5e**) and H-UAPGEL-PAM (**5f**) exhibited interesting features, as shown in Figure 3c, where the estimated second-order rate constants (*k*_2_) are graphically compared (see also Appendix A). The catalytic activity was significantly increased by introducing basic K, while the activity remained as low as those of **5c** and **5d** when neutral P was introduced instead of H. Furthermore, no activity was observed for **5g**, which does not have U, clearly indicating that U is essential for catalytic activity. The reaction was also monitored in the case of **5e** by ^1^H NMR in CD_3_OD (Figure 3d). The oxidation of DTT^red^ to DTT^ox^ was clearly observed with the reaction time. In this case, the reaction proceeded faster than that observed in the UV assay, due to higher concentrations of H_2_O_2_ and DTT^red^. Resin **5e**, which was recovered from the reaction mixture by filtration, exhibited GPx-like peroxidase activity at least two more times. However, the catalytic efficiency gradually decreased, probably due to overoxidation of the selenium moiety by H_2_O_2_ (Appendix A). Thus, on-resin selenopeptide **5** was found to be reusable as a GPx-like antioxidant catalyst.

It is sometimes reported in the literature that the introduction of a basic amino group in the proximity of the Se atom enhances the GPx-like peroxidase activity of organoselenium compounds [53,54,55,56,57]. However, the mechanism for activity enhancement has not been well understood. We recently proposed that the NH···Se hydrogen bond formed between U and a proximate basic amino acid residue would activate the selenosulfide bond (Se-S) of the stable selenenyl sulfide intermediate (RSe-SR’) [14,18,58]. To investigate the possible formation of an NH···Se hydrogen bond between the Se atom of U and the sidechain ammonium group of K for **5e**, replica exchange Monte Carlo molecular simulation using the three-dimensional single-amino acid potential force field (REMC/SAAP3D) [59] was performed for H-UAKGEL-OH. The clustering analysis of the output structures at 300 K revealed that the structure with the (N)H···Se atomic distance between the Se atom of U and -NH_3_^+^ ammonium group of K less than 3 or 4 Å was populated in 10 or 26%, respectively, in water (Appendix A). The representative molecular structure is shown in Figure 4a. The results strongly suggested the formation of a similar NH···Se hydrogen bond in the selenenyl sulfide intermediate state (RSeSR’) (Figure 4b), in which a nucleophilic attack of the second thiol at the S atom of the Se-S bond to liberate DTT^ox^ and RSe^−^ is accelerated by the NH···Se hydrogen bond. In another scenario, the basic amino group might deprotonate a thiol substrate to enhance nucleophilicity, thereby accelerating the catalytic cycle [60]. However, this is less feasible in this case, because the amino group that had been protonated by the treatment with TFA during the deprotection process did not have a capacity to accept a proton from the thiol substrate.

### 2.3. Oxidative Cyclization of β,γ-Unsaturated Acids Catalyzed by ***6***

The catalytic activity of on-resin selenopeptides in the conversion of β,γ-unsaturated acids (**7**) into α,β-unsaturated lactones (**8**) was assessed using resin **6**. For this type of selenium-catalyzed oxidative cyclization reaction, diaryl diselenides (ArSeSeAr) are usually used as catalysts in the presence of excess ammonium peroxodisulfate ((NH_4_)_2_S_2_O_8_) as an oxidant [61]. If aliphatic diselenides (RSeSeR) were employed instead, the β elimination of the oxidized selenium moiety would take place in the direction of the alkyl group (R) during the reaction cycle, hence the selenium compounds should lose catalytic activity [62]. We previously reported that α-methyl-L-selenocystine derivatives ((X-MeSec-OR)_2_; X = Ac or Boc, R = Me, Et, etc.) that do not have an eliminable hydrogen atom at the α-carbon atom can be utilized as an asymmetric catalyst for the cyclization reaction of **7** [19]. However, neither the reaction yields (12–84%) nor asymmetric yields (7–27% e.e.) were satisfactory for practical applications. In this study, we hypothesized that a Sec residue on a resin, even though it possesses an eliminable H atom, can be applied as a catalyst, because the β elimination would be hindered by a steric effect of the bulky polymer support. Our preliminary examination of the reaction conditions indicated that cyclized product **8** was only obtained when on-resin selenopeptides that consist of only non-polar amino acid residues, with protection of the N-terminal amino group, were employed. Thus, **6a**–**d**, with acetyl or Fmoc capping of the N-terminus, were synthesized and employed for the cyclization reaction of **7** as a catalyst. The results are summarized in Table 2. It should be noted that it was not necessary to deprotect the PMB group on the Se atom of U, because it can be removed in situ under the oxidative conditions [63].

First, (*E*)-4-phenylpent-3-enoic acid (**7a**) was employed as a substrate. The reaction was carried out in acetonitrile, using (NH_4_)_2_S_2_O_8_ as an oxidant, under the reaction conditions reported in the literature [61]. When resin **6a** or **6b** were employed as a catalyst, the desired product **8a** was obtained from **7a** in a yield of 51 or 25%, respectively, under the diluted condition, i.e., in 10 mL MeCN at 60 °C for 18 h (entries 1 and 2). To enhance the reaction efficiency, the volume of the solvent was then reduced from 10 to 1 mL. Under this high concentration condition, the yields of **8a** were increased to 79% and 33% for catalysts **6a** and **6b**, respectively, as expected (entries 3 and 4, round 1). It is of note that **6a**, with a short peptide spacer between U and the resin base, was more efficient as a catalyst than **6b,** with a long peptide spacer. Furthermore, 1,4-Dioxane was also usable as a solvent, but the reaction proceeded slowly compared to that in acetonitrile (entry 5, round 1). Similarly, when catalyst **6c** with an Fmoc capping on the N-terminus was employed, **8a** was obtained in 47% yield (entry 6). Interestingly, the yield of **8a** was significantly increased to 90% when resin **6d**, in which the N-terminal U and A residues of **6a** were exchanged, was employed as a catalyst (entry 7, round 1). In most cases, product **8a** was obtained as a racemic mixture (% e.e. <10) (Appendix A), suggesting that further design or tuning of the selenopeptides on the resin is required to enhance the asymmetric induction.

Next, the scope of substrate **7** was examined using catalyst **6a** under the optimal reaction conditions, i.e., in acetonitrile (1 mL) at 60 °C for 18 or 48 h (Table 2, entries 8–11). The reaction proceeded smoothly for **7b**, in which the Me group of **7a** is replaced by H, to produce **8b** in 53% yield. However, when substrate **7c,** with a *p*-methoxy substituent on the phenyl ring was applied, the desired product **8c** was not yielded, even after prolonged reaction time. On the other hand, the reaction proceeded efficiently for aliphatic alkenes **7d** and **7e**. The results indicated that on-resin selenopeptides **6** are useful as a catalyst for both aromatic and aliphatic unsaturated acids (**7**), but there is some limitation in the substrate scope.

### 2.4. Recycling of On-Resin Selenopeptide Catalysts (***6***)

Reusability of on-resin selenopeptides **6a**, **6b**, and **6d** as catalysts for the oxidative cyclization of **7** to **8** was subsequently investigated. The resin was recovered from the reaction mixture by filtration and reused under the same reaction conditions repeatedly. The results clearly demonstrated that the selenopeptides can be used as a catalyst two or three times (Table 2), although the yields of **8a** decreased with the recycling rounds, accompanying color change from pale yellow to red (Figure 5a). To assess the reason for the observed deterioration, the peptide was cleaved from the used resin. However, no visible peptide was obtained, suggesting the degradation of the selenopeptide on the resin. Indeed, microscopic investigation of resin **6d** using the electron probe micro analyzer (EPMA) revealed that selenium had been detached from the resin after recycling (round 3), while the shape appearance of the resin did not change significantly, maintaining clear signals due to O and C elements (Figure 5b,c).

### 2.5. Mechanism for the Conversion from ***7*** to ***8*** Catalyzed by On-Resin Selenopeptides (***6***)

The proposed reaction mechanism is shown in Figure 6. The PMB protecting group on the Sec residue is removed via reaction with a peroxodisulfate anion (S_2_O_8_^2−^), probably through the selenoxide or selenonium intermediate, to generate the electrophilic selenium species A, which subsequently reacts with substrate **7** to produce the seleniranium intermediate B. Intramolecular nucleophilic attack of a carboxylate to the γ-position produces the cyclized intermediate C, which finally releases lactone **8** and regenerates the active species A via reaction with excess S_2_O_8_^2−^. However, in the last step, the abstraction of the α-H atom of Sec also takes place slowly to degrade the Sec residue, leaving a dehydroalanine moiety on the resin. This degradation path explains the observed deactivation of catalyst **6**.

## 3. Materials and Methods

### 3.1. Materials

Boc-Leu-PAM and Boc-Gly-PAM resins were purchased from Merck KGaA (Darmstadt, Germany). Fmoc-Sec(PMB)-OH was synthesized as previously reported [64]. (*E*)-4-phenylpent-3-enoic acid (**7a**), (*E*)-4-(4-methoxyphenyl)but-3-enoic acid (**7c**), and (*E*)-5-phenylpent-3-enoic acid (**7d**) were synthesized according to the literature methods [63,65]. All other reagents and substrates were commercially available from domestic suppliers and used without further purification.

### 3.2. General Procedures for the Synthesis of On-Resin Selenopeptide Catalysts

All peptides were grown on the PAM resin by applying the 9-fluorenylmethoxycarbonyl (Fmoc)-based solid-phase method. Completion of the amino acid coupling was confirmed using Kaiser test.

PAM resin (40.0 mg, 25 μmol) was weighed in a plastic reaction vessel, in which the resin was swelled with dichloromethane (DCM) for 2 h at room temperature. The resin was then treated with trifluoroacetic acid (TFA) for 10 s, with vortex mixing. The deprotection reaction was repeated in TFA for 2 min. The resin was then washed with DCM (×5) and neutralized with 5% diisoprorylethylamine (DIEA)/DCM for 5 min twice. After washing with *N*,*N*-dimethylformamide (DMF) for 1 min (×5), Fmoc-protected amino acid (100 μmol), which was pre-activated with 1 M 1-hydroxybenzotriazole (HOBt)/DMF (150 μL) and 1 M *N*,*N’*-dicyclohexylcarbodiimide (DCC)/DMF (150 μL) for 30 min at room temperature, was added to the resin. The mixture was vortexed for 60 min at 50 °C. After the coupling, the resin was washed with DMF for 5 min, 50% MeOH/DCM for 1 min, and then DMF for 1 min (×2). The resin was treated with 10% acetic anhydride (Ac_2_O)/5% DIEA/DMF (1 mL) for 5 min to cap the unreacted N-terminus. After washing with DMF for 1 min (×5), the resin was treated with 20% piperidine/DMF (1 mL) for 5 min, and the piperidine treatment was repeated for 15 min. The same coupling procedure was applied to subsequent amino acids for growing a peptide chain on the resin. To couple a Sec residue to the peptide, Fmoc-Sec(PMB)-OH (25.5 mg, 50 μmol) was activated with 1 M HOBt/DMF (75 μL) and *N*,*N’*-diisopropylcarbodiimide (DIPCI) (11.6 μL, 75 μmol) for 30 min at room temperature, and added to the resin. The mixture was vortexed for 30 min at 50 °C. The mixture was again added with DIPCI (11.6 μL, 75 μmol) and vortexed for 60 min at 50 °C to complete the coupling. The resin was finally treated with 20% piperidine/DMF, as described above.

For the synthesis of **5**, the resin (40 mg) thus obtained was treated with a TFA cocktail (TFA:TIS:H_2_O:DPDS = 90:2.5:2.5:5) (300 μL) for 2 h at room temperature, to deprotect all protecting groups, including PMB, on the sidechains. After washing with DMF for 1 min (×3), the resin was treated with 2 mM DTT^red^/MeOH (1 mL) for 5 min at room temperature (×3). The resin was washed with 50% MeOH/CDM for 1 min (×3) and then DCM for 1 min (×3). The obtained resin was dried in vacuo to yield on-resin selenopeptide **5**.

For the synthesis of **6**, the obtained resin (40 mg) was acetylated with 10% Ac_2_O/5% DIEA/DMF (1 mL) for 5 min at room temperature and then washed with DMF for 1 min (×3), 50% MeOH/DCM for 1 min (×3), and DCM for 1 min (×3). The resulting resin was dried in vacuo.

### 3.3. Characterization of Selenopeptides

The resin (**5**, **6**, or the synthetic precursors) (20 mg), thioanisol (50 μL), and 1,2-ethanedithiol (EDT) (25 μL) were weighed in an Eppendorf tube, and the mixture was cooled on an ice bath. The mixture was added with TFA (500 μL). After stirring for 10 min on the ice bath, the mixture was added with trifluoromethanesulfonic acid (TfOH) (50 μL) and then stirred for 1 h at room temperature. The mixture was filtered, and the filtrate was added with cooled diethyl ether (Et_2_O). The precipitates were collected by centrifugation at 6000 rpm for 5 min. After washing with Et_2_O (×6), the obtained peptide was dried in vacuo for 2 h. The resulting peptide was analyzed using HPLC, MALDI-TOF-MS, and amino acid analysis to confirm the purity, identity, and reaction yield.

HPLC analysis was performed on a 10A series HPLC system (Shimadzu Co., Kyoto, Japan), equipped with a 1 mL sample solution loop and a TSKgel ODS-100V 4.6 × 150 RP-column (Tosoh Co., Tokyo, Japan). The sample solution was prepared by dissolving the peptide, which was cleaved from the resin, in 50% acetonitrile/H_2_O containing 0.1% TFA (100 μL). The solvent gradient was applied with 0.1% TFA in water (eluent A) and 0.1% TFA in acetonitrile (eluent B). In the analysis for the peptides cleaved from **5a**–**g**, the column was initially equilibrated with 10% B at a flow rate of 0.5 mL/min. After injection of the sample solution (4–10 μL), the concentration of B was linearly increased from 10% to 35% in 23 min. On the other hand, the solvent gradient of 0 to 100% B in 23 min was applied for the peptides cleaved from **6b** and the precursors of **6a** and **6d**. In the case of **6c**, the analysis was not performed, because **6c** was the synthetic precursor of **6a**. The peptide samples fractionated were collected, lyophilized, and analyzed using MALDI-TOF-MS and amino acid analysis. MALDI-TOF-MS spectra were measured on a JMS-S3000 mass spectrometer (JEOL Ltd., Akishima, Japan) by using α-cyano-4-hydroxycinnamic acid as a matrix. Amino acid analysis was conducted by hydrolyzing the sample in concentrated hydrochloric acid for 120 min at 150 °C.

H-UHGEL-OH cleaved from resin **5a** was mainly obtained as an oxidized diselenide form. Yield, 24%. MALDI-TOF-MS (*m*/*z*) found: 1209.43, calcd for C_44_H_69_N_14_O_16_Se_2_^+^ [M+H]^+^: 1209.33. Amino acid analysis: Glu_1.06_Gly_1.00_His_0.80_Lue_0.87_.

(H-UAHGEL-OH)_2_ cleaved from **5b**. Obtained as an oxidized diselenide form. Yield, 49%. MALDI-TOF-MS (*m*/*z*) found: 1351.63, calcd for C_50_H_79_N_16_O_18_Se_2_^+^ [M+H]^+^: 1351.41. Amino acid analysis: Ala_1.00_Glu_1.25_Gly_1.00_His_1.01_Lue_1.00_.

(H-UAAAHGEL-OH)_2_ cleaved from **5c**. Obtained as an oxidized diselenide form. Yield, 49%. MALDI-TOF-MS (*m*/*z*) found: 1635.53, calcd for C_62_H_99_N_20_O_22_Se_2_^+^ [M+H]^+^: 1635.56. Amino acid analysis: Ala_3.00_Glu_1.34_Gly_1.00_His_1.20_Lue_1.03_.

(H-UAAAAHGEL-OH)_2_ cleaved from **5d**. Obtained as an oxidized diselenide form. Yield, 23%. MALDI-TOF-MS (*m*/*z*) found: 1777.68, calcd for C_68_H_109_N_22_O_24_Se_2_^+^ [M+H]^+^: 1777.63. Amino acid analysis: Ala_4.00_Glu_1.74_Gly_0.98_His_0.77_Lue_0.96_.

(H-UAKGEL-OH)_2_ cleaved from **5e**. Obtained as an oxidized diselenide form. Yield, 51%. MALDI-TOF-MS (*m*/*z*) found: 1333.49, calcd for C_50_H_89_N_14_O_18_Se_2_^+^ [M+H]^+^: 1333.48. Amino acid analysis: Ala_1.00_Glu_1.05_Gly_1.00_Lys_0.99_Lue_1.31_.

(H-UAPGEL-OH)_2_ cleaved from **5f**. Obtained as an oxidized diselenide form. Yield, 22%. MALDI-TOF-MS (*m*/*z*) found: 1271.46, calcd for C_48_H_79_N_12_O_18_Se_2_^+^ [M+H]^+^: 1271.42. Amino acid analysis: Ala_1.00_Glu_1.02_Gly_1.01_Leu_1.00_Pro_1.17_.

H-AAHGEL-OH cleaved from **5g**. Yield, 57%. MALDI-TOF-MS (*m*/*z*) found: 597.37, calcd for C_25_H_41_N_8_O_9_^+^ [M+H]^+^: 597.30. Amino acid analysis: Ala_2.00_Glu_0.92_Gly_1.00_His_0.98_Lue_0.98_.

(H-UAAAG-OH)_2_ cleaved from H-U*AAAG-PAM, a synthetic precursor of **6a**. Obtained as an oxidized diselenide form. Yield, 41%. MALDI-TOF-MS (*m*/*z*) found: 899.21, calcd for C_28_H_48_N_10_NaO_12_Se_2_^+^ [M+Na]^+^: 899.17. Amino acid analysis: Ala_3.00_Gly_1.00_.

(Ac-UAAAGGG-OH)_2_ cleaved from **6b**. Obtained as an oxidized diselenide form. Yield, 11%. MALDI-TOF-MS (*m*/*z*) found: 1211.30, calcd for C_40_H_64_N_14_NaO_18_Se_2_^+^ [M+Na]^+^: 1211.27. Amino acid analysis: Ala_3.00_Gly_3.22_.

(H-AUAAG-OH)_2_ cleaved from Fmoc-AU*AAG-PAM, a synthetic precursor of **6d**, after Fmoc deprotection. Obtained as an oxidized diselenide form. Yield, 26%. MALDI-TOF-MS (*m*/*z*) found: 877.20, calcd for C_28_H_49_N_10_O_12_Se_2_^+^ [M+H]^+^: 877.19. Amino acid analysis: Ala_0.99_Gly_3.00_.

### 3.4. Assessment of Peroxidase Activity

The GPx-like peroxidase activity of **5a**–**g** was evaluated using two methods, using the reaction between H_2_O_2_ and DTT^red^ in methanol at room temperature. In the UV method, DTT^red^ (15.4 mg, 0.10 mmol) and catalyst **5** (10 mg, 6.0 μmol, 0.06 eq) were weighed in a 10 mL round-bottom flask. Methanol (10 mL) was added to the flask, and the mixture was gently stirred. To the resulting mixture, 44% H_2_O_2_ (8.0 μL, 0.10 mmol) was added, with continuous stirring. An aliquot of the solution (900 μL) was taken out hourly, and the UV absorbance at 310 nm was measured for 6 h. After each UV measurement, the sample solution was recovered in the reaction flask. The experiment was repeated at least three times. The maximum change in the absorbance at 310 nm was determined to be 1.20 after 18 h, when DTT^red^ was completely oxidized to DTT^ox^. Thus, the difference in the molar extinction coefficients at 310 nm between DTT^red^ and DTT^ox^ (Δ*ε*) was estimated to be 120 M^−1^cm^−1^, which was consistent with the literature [66]. In the NMR method, DTT^red^ (22.0 mg, 0.14 mmol) and catalyst **5e** (10 mg, 6.0 μmol, 0.043 eq) were weighed in an Eppendorf tube. CD_3_OD (1.0 mL) was added to the tube, and the mixture was gently stirred. To the resulting mixture, 44% H_2_O_2_ (11 μL, 0.14 mmol) was added. The ^1^H NMR spectrum of the solution part was recorded hourly on an AV-500 spectrometer (Bruker Co., Billerica, MA, US) at 500 MHz by spinning down the resin before taking the sample solution (500 μL) into the NMR tube. After the NMR measurement, the solution was recovered in the reaction vessel, and stirring of the mixture continued.

### 3.5. Kinetic Analysis

The second-order rate constant (*k*_2_) for the reaction between H_2_O_2_ and DTT^red^ in methanol in the presence of catalyst **5** was determined based on the observed absorbance change at 310 nm. Since the reaction rate *v* can be expressed by the equation, *v* = –d[DTT^red^]/d*t* = *k*_2_[H_2_O_2_][DTT^red^] = *k*_2_[DTT^red^]^2^, the reciprocal of [DTT^red^] was plotted against the reaction time. The slope of the plots gave the *k*_2_ value.

### 3.6. Statistical Analysis

The values were expressed using means and standard deviations (SD). The one-way analysis of variance (ANOVA) was used to determine whether there are any statistically significant differences among the means of independent groups. Tukey’s test was performed as a post hoc analysis to address the pairs of groups that have significant differences. When *p* < 0.05, the differences were regarded as statistically significant. All statistical analyses were carried out using the Microsoft Excel program version 2412, using built-in functions for one-way ANOVA. For the post hoc test, the analysis was manually conducted using the same program.

### 3.7. Catalytic Cyclization of β,γ-Unsaturated Acids (***7***)

In a general procedure, catalyst **6** (10 mg, 6.0 μmol, 0.026 eq), substrate **7a** (40.0 mg, 0.23 mmol), and ammonium peroxodisulfate ((NH_4_)_2_S_2_O_8_) (155 mg, 0.68 mmol, 3 eq) were weighed in a 10 mL round-bottom flask. Acetonitrile (1 mL) was added to the flask. The mixture was vigorously stirred at 60 °C for 18 h. After suction filtration of the reaction mixture, the filtrate was evaporated and the residue was purified by silica gel column chromatography (ethyl acetate–hexane) to isolate product **8a**. The absolute stereochemistry for the major enantiomer of **8a** was determined by HPLC using a chiral column (OD-RH 0.46 cmf × 15 cm column; Daicel Co., Osaka, Japan) according to the reported method [19]. The reference sample was synthesized following the literature method [67]. On the other hand, the resin and excess (NH_4_)_2_S_2_O_8_ collected via suction filtration were washed with H_2_O and then acetonitrile. The recovered resin was stored in the same solvent and then reused. Cyclization products **8a**, **8b**, **8d**, and **8e** were characterized by measuring the ^1^H NMR spectra, which were found to be consistent with those reported in the literature [63,68].

### 3.8. EPMA Analysis

The microscopic images and characteristic X-ray spectra of sample resins were measured on an EPMA-8050G Electron Probe Micro Analyzer (EPMA) (Shimadzu Co., Kyoto, Japan) using pentaerythritol (PET), ammonium dihydrogen phosphate (ADP), lithium fluoride (LiF), layered structure analyzer (LSA70), and rubidium acid phthalate (RAP) as analyzing crystals. Boc-Gly-PAM resin, **6d** before use, and **6d** after use, which were well dried under a vacuum, were employed as samples for investigation. The samples were deposited on a sample holder using sticky carbon tape. Measurement conditions were accelerating voltage: 15.0 (kV); beam current: 5.0 (nA); beam size: 1 µm; and integration time: 90.0 (ms/point).

### 3.9. Molecular Simulation

The molecular structures for **5e** were calculated by replica-exchange Monte Carlo molecular simulation, using the single amino acid potential force field (REMC/SAAP3D) [69,70]. The simulation was performed for the truncated model of **5e**, i.e., H-UAKGEL-OH, applying a selenolate form for the sidechain of U, a protonated ammonium form for the sidechain of K, and a carboxylate form for the sidechain of E. Simulation conditions were set to the same as the previous work [59]: the number of replicas 4, temperature 300, 370, 440, and 510 K, Monte Carlo steps 4 × 10^8^, output structures 2 × 10^4^, and frequency of the replica exchange every 2 × 10^4^ MC steps, applying a sigmoidal distance-dependent dielectric model. The output structures obtained at 300 K were classified into 20 structural clusters using the *k*-means method, applying the all-atom RMSD values [59]. The graphics were drawn using ViewerLite 5.0 (Accelrys Inc., San Diego, CA, US).

## 4. Conclusions

We succeeded in the development of new polymer-supported selenopeptide catalysts (**5** and **6**). The synthesized on-resin selenopeptide catalyst **5** was usable as a GPx-like antioxidant catalyst (Figure 3). The catalytic activity could be enhanced by arranging a basic amino acid close to the Sec residue. In particular, **5e** having a Lys residue showed high peroxidase activity. On the other hand, on-resin catalyst **6** was successfully applied to the catalytic cyclization of unsaturated acid **7** to **8** (Table 2), although the asymmetric yields were low. In both reactions, the on-resin catalysts were found to be reusable. Since selenocysteine (U) is a natural amino acid, on-resin selenopeptides are potential targets as environmentally friendly green redox catalysts. However, the current on-resin catalysts gradually lost their activities with recycling rounds. Therefore, to enhance the chemical stability of selenopeptides on a resin, remodeling the amino acid sequence, as well as selection of the proper resin base, is necessary in future studies. It is also an interesting challenge to apply such catalysts to biocompatible materials [71] that mimic the functions of selenoenzymes—not only GPx, but also other selenoenzymes—as well as to the catalytic conversion of β-substituted-β,γ-unsaturated acids [72], which would significantly enhance the usability of the on-resin selenopeptide catalysts.

## Figures and Tables

**Figure 1 molecules-30-00480-f001:**
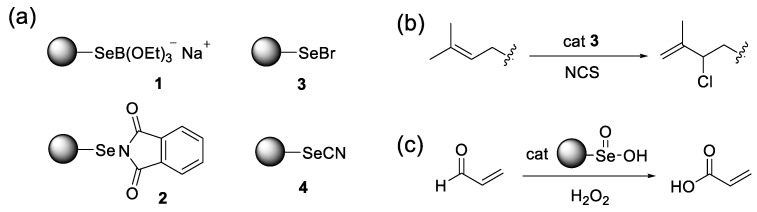
Polymer-bound selenium reagents. (**a**) Representative nucleophilic or electrophilic selenium reagents. (**b**) Catalytic chlorination of alkenes using polystyrene-bound selenenyl bromide (**3**) [47]. (**c**) Oxidation of acrolein with H_2_O_2_ catalyzed by selenium-modified microgel [48].

**Figure 2 molecules-30-00480-f002:**
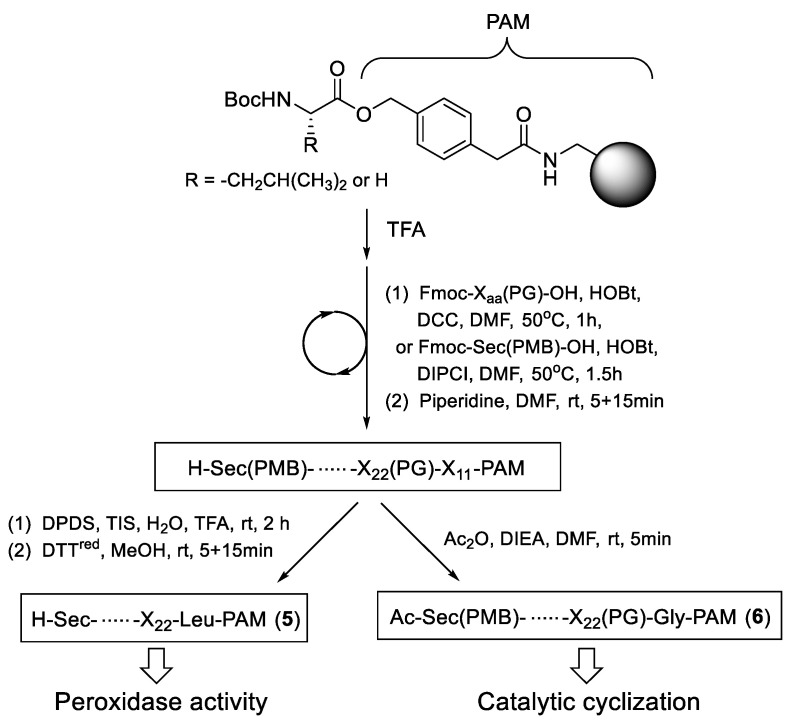
Synthesis of on-resin selenopeptide catalysts **5** and **6**.

**Figure 3 molecules-30-00480-f003:**
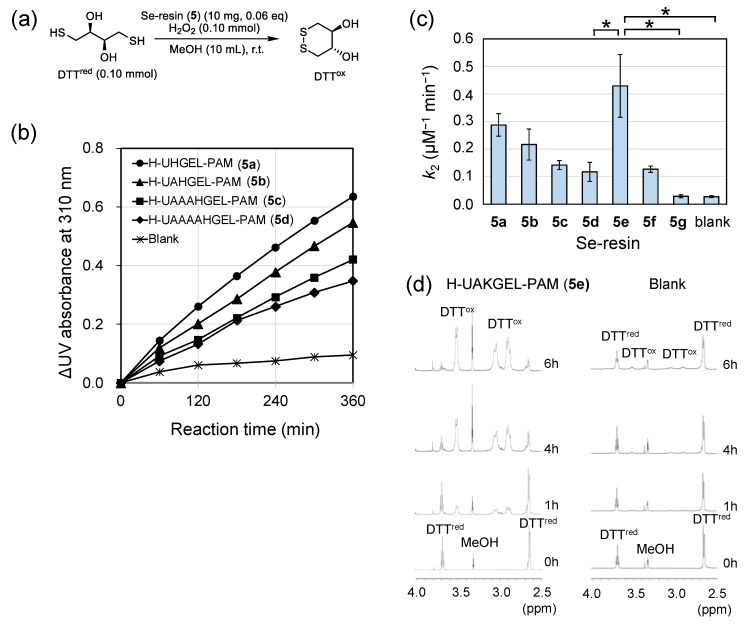
Assessment of the GPx-like peroxidase activity of **5**. (**a**) The reaction conditions applied for the UV assay. (**b**) Changes in the UV absorbance of the reaction solution at 310 nm, with the reaction time in the presence of **5a**–**d**. (**c**) The second-order rate constants (*k*_2_) determined for **5a**–**g**. Bars are shown as mean ± SD (*n* = 3). Asterisk represents a significant difference between the groups by one-way ANOVA and the post hoc Tukey test (* *p* < 0.05). (**d**) The changes in ^1^H NMR spectra with the reaction time. Conditions: Se-resin (**5e**) 10 mg, [DTT^red^] = [H_2_O_2_] = 140 mM in CD_3_OD (1.0 mL) at 300 K.

**Figure 4 molecules-30-00480-f004:**
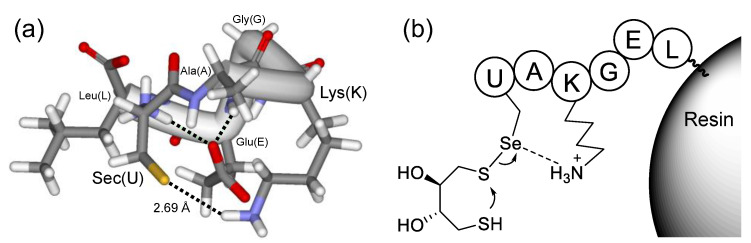
Involvement of an NH···Se hydrogen bond in the catalytic cycle of **5e**. (**a**) The representative structure (cluster 5, 7.8%) obtained by REMC/SAAP3D molecular simulation at 300 K in water for H-UAKGEL-OH. Hydrogen bonds are indicated by dotted lines. (**b**) A possible function of the NH···Se hydrogen bond formed between U and K residues in the selenenyl sulfide intermediate.

**Figure 5 molecules-30-00480-f005:**
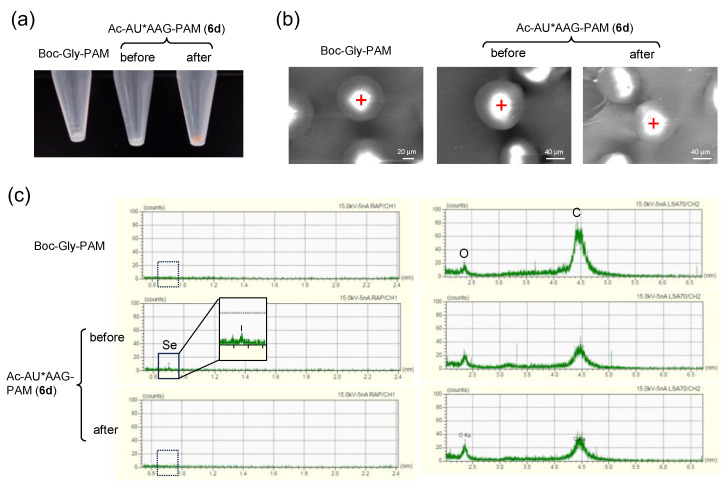
Appearance and EPMA investigation of resin **6d**. (**a**) Appearance of the resin base (Boc-Gly-PAM) **6d** before use, and **6d** after use. (**b**) Microscopic images of the resins with irradiation of electron beam of 5 nA. Red crosses indicate irradiation points. (**c**) Characteristic X-ray spectra observed for the resins by EPMA, with beam size 1 μm. A solid box shows that the signal corresponding to a Se atom was present at 0.90 nm for **6d** before use, but it was not observed after use, like Boc-Gly-PAM, as shown by dotted boxes (a left column). In the meantime, the signals corresponding to C and O atoms at 4.47 and 2.37 nm, respectively, were clearly observed for all samples (a right column).

**Figure 6 molecules-30-00480-f006:**
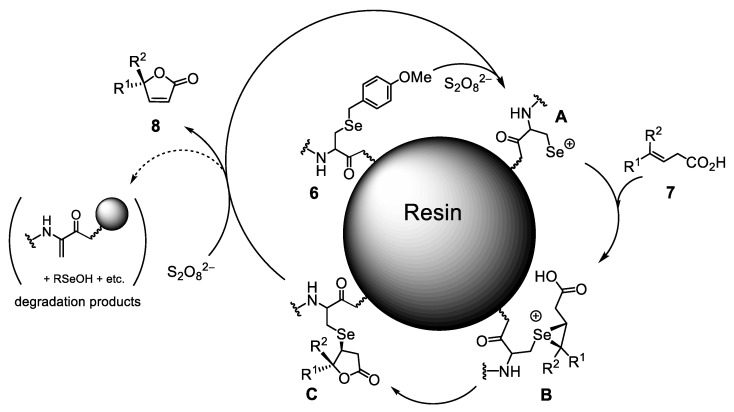
A plausible mechanism for the oxidative cyclization of **7** catalyzed by **6**.

**Table 1 molecules-30-00480-t001:** On-resin selenopeptide catalysts (**5** and **6**) synthesized in this study.

Resins	Amino Acid Sequence ^1^	Resins	Amino Acid Sequence ^1^
**5a**	H-UHGEL-PAM	**6a**	Ac-U*AAAG-PAM
**5b**	H-UAHGEL-PAM	**6b**	Ac-U*AAAGGG-PAM
**5c**	H-UAAAHGEL-PAM	**6c**	Fmoc-U*AAAG-PAM
**5d**	H-UAAAAHGEL-PAM	**6d**	Ac-AU*AAG-PAM
**5e**	H-UAKGEL-PAM		
**5f**	H-UAPGEL-PAM		
**5** **g**	H-AAHGEL-PAM		

^1^ U: selenocysteine (Sec); U*: Sec (PMB); PAM: (4-hydroxymethylphenyl)acetamidomethyl polystyrene.

**Table 2 molecules-30-00480-t002:** Cyclization of β,γ-unsaturated acids (**7**) into α,β-unsaturated lactones (**8**) catalyzed by on-resin selenopeptides (**6**) ^1^.

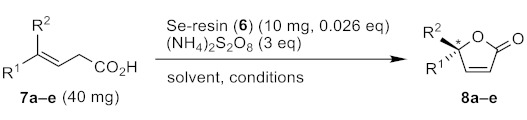						
Entry	Substrate	Se-Resin	Solvent	Conditions	Round	Yields of 8
1	**7a** (R^1^ = Ph, R^2^ = Me)	**6a**	MeCN (10 mL)	60 °C, 18 h	1	51%
2	**7a**	**6b**	MeCN (10 mL)	60 °C, 18 h	1	25%
3	**7a**	**6a**	MeCN (1 mL)	60 °C, 18 h	1 2 3 4	79% 73% 27% ^2^ 0%
4	**7a**	**6b**	MeCN (1 mL)	60 °C, 18 h	1 2 3 4	33% 51% 14% ^2^ 0%
5	**7a**	**6b**	1,4-dioxane (1 mL)	60 °C, 48 h	1 2 3	33% 18% 0%
6	**7a**	**6c**	MeCN (1 mL)	60 °C, 18 h	1	47%
7	**7a**	**6d**	MeCN (1 mL)	60 °C, 18 h	1 2 3	90% 83% trace
8	**7b** (R^1^ = Ph, R^2^ = H)	**6a**	MeCN (1 mL)	60 °C, 18 h	1	53%
9	**7c** (R^1^ = 4-(MeO)Ph, R^2^ = H)	**6a**	MeCN (1 mL)	60 °C, 48 h	1	0%
10	**7d** (R^1^ = CH_2_Ph, R^2^ = H)	**6a**	MeCN (1 mL)	60 °C, 18 h	1	45%
11	**7e** (R^1^ = Et, R^2^ = H)	**6a**	MeCN (1 mL)	60 °C, 18 h	1	68%

^1^ The asymmetric carbon atom of **8** is labelled with an asterisk. ^2^ Product **8** was obtained as a mixture with an unknown byproduct, which could not be separated by column chromatography.

## Data Availability

The original contributions presented in the study are included in the article and Appendix A. Further inquiries can be directed to the corresponding author.

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
