# Peer review of "On-Resin Selenopeptide Catalysts: Synthesis and Applications of Enzyme-Mimetic Reactions and Cyclization of Unsaturated Carboxylic Acidsâ€"

_molecules, 2025, doi:10.3390/molecules30030480_

Round 1
Reviewer 1 Report
Comments and Suggestions for Authors
The manuscript comprehensively explores the use of on-resin selenopeptide catalysts for mimicking enzyme activity, specifically glutathione peroxidase-like activity, and their application in oxidative cyclization reactions. This innovative approach aligns with the contemporary goals of sustainable catalysis by utilizing recyclable, environmentally friendly selenium-containing peptides.
This work excels in mechanistic elucidation, such as the example of NH···Se hydrogen bonding as a new and rarely explored interaction in catalysis. The demonstration of dual functionality of these catalysts across two distinct reaction types adds enormous significance to this work.
The synthesis and characterization of these selenopeptides are explained methodically. Their results are strongly backed up by analyses such as HPLC, MALDI-TOF-MS, and EPMA. Furthermore, the reported catalytic and reusability results agree well with the stated conclusions, further improving the reliability. However, the manuscript can be accepted for publication after this correction:
(1) Provide a comparison table of previously reported selenopeptide catalysts in tabular form
(2) Page 6 line 194, read the sentence carefully and correct the word “longer spacer”
Author Response
We thank the reviewer for careful reading of our manuscript and providing valuable comments, which are useful to improve the manuscript. We corrected the manuscript accordingly.
Comment 1: Provide a comparison table of previously reported selenopeptide catalysts in tabular form
Response: We made a requested table as Table S2. It should be noted, however, that the direct comparison of the catalytic activities between on-resin catalysts and previously reported selenopeptide catalysts is not possible because on-resin selenopeptides were employed in methanol in this study as heterogeneous catalysts, while previously reported selenopeptide catalysts were employed in aqueous buffer solution as homogeneous catalysts.
Comment 2: Page 6 line 194, read the sentence carefully and correct the word “longer spacer”
Response: We corrected the sentence as follows: It is of note that 6a with a short peptide spacer between U and the resin base was more efficient as a catalyst than 6b with a long peptide spacer.
Reviewer 2 Report
Comments and Suggestions for Authors
Iwaoka and coworkers report a very interesting study on supported Se-peptides as competent and effective catalysts for promoting enzime-mimetic transformations and cyclization of unsaturated carboxyls. After a comprehensive and well-balanced description of the state-of-the-art of Se-mediated processes - highlighting the most recent advancements on polymer-bound Se-reagents - they embarked in an effective preparation of two catalysts: notably, experimental details are discussed both in the manuscript and in the Supp Info. Therefore, the catalytic activities are documented towards GPx-like peroxidase (furnishing di-hydroxylated cyclic sulfides) and towards the cyclization of unsaturated carboxylic acids (giving valuable unsaturated lactones). In particular, this second transformation has been investigated in terms of scope . Authors evidence the difficulties in achieving high level of stereocontrol. It is importnat to observe the recyclability of the catalysts and the convincing rationales provided for both reactions.
This reviewer finds the study highly suitable for the publication in Molecules. However, prior to the final acceptance would recommend to comment the suitability of the catalyst for engaging as a competent substrate a carboxylic acid presenting a substituent at the beta-position of the acid. This would certainly make more appealing the overall study.
Author Response
Comment: This reviewer finds the study highly suitable for the publication in Molecules. However, prior to the final acceptance would recommend to comment the suitability of the catalyst for engaging as a competent substrate a carboxylic acid presenting a substituent at the beta-position of the acid. This would certainly make more appealing the overall study.
Repsonse: We thank the reviewer for careful reading of our manuscript and providing valuable comments, which are useful to improve the manuscript. We added the relevant comment at the end of Conclusion: "as well as to the catalytic conversion of β-substituted-β,γ-unsaturated acids [72], which would significantly enhance the usability of the on-resin selenopeptide catalysts". A new reference [72] is now cited.